# MACH: Embarrassingly parallel $K$-class classification in $O(d \log K)$ memory and $O(K \log K + d \log K)$ time, instead of $O(Kd)$

## Abstract

We present Merged-Averaged Classifiers via Hashing (MACH) for $K$-classification with large $K$. Compared to traditional one-vs-all classifiers that require $O(Kd)$ memory and inference cost, MACH only need $O(d \log K)$ memory while only requiring $O(K \log K + d \log K)$ operation for inference. MACH is the first generic $K$-classification algorithm, with provably theoretical guarantees, which requires $O(\log K)$ memory without any assumption on the relationship between classes. MACH uses universal hashing to reduce classification with a large number of classes to few independent classification task with small (constant) number of classes. We provide theoretical quantification of accuracy-memory tradeoff by showing the first connection between extreme classification and heavy hitters. With MACH we can train ODP dataset with 100,000 classes and 400,000 features on a single Titan X GPU (12GB), with the classification accuracy of 19.28%, which is the best-reported accuracy on this dataset. Before this work, the best performing baseline is a one-vs-all classifier that requires 40 billion parameters (160 GB model size) and achieves 9% accuracy. In contrast, MACH can achieve 9% accuracy with 480x reduction in the model size (of mere 0.3GB). With MACH, we also demonstrate complete training of fine-grained imagenet dataset (compressed size 104GB), with 21,000 classes, on a single GPU.

## 1 Motivation

The area of extreme multi-class classification has gained significant interest recently (Choromanska & Langford, 2015). Extreme multi-class refers to the vanilla multi-class classification problem where the number of classes $K$ is significantly large. A large number of classes $K$ brings a new set of computational and memory challenges in training and deploying classifiers.

The growth in the volume and dimensionality of data is a well-known phenomenon. Concurrently, there is also a significant growth in the number of classes or labels of interest. There are several reasons behind this growth. In the last decade, it has been shown that many hard AI problems can be naturally modeled as a massive multi-class problem, leading to a drastic improvement over prior art. The success in NLP is a classic example of this phenomena. For example, popular models predict the best word, given the full context observed so far. Such models are becoming the state-of-the-art in machine translation (Sutskever et al., 2014), word embeddings (Mikolov et al., 2013), etc. For large-dataset, the vocabulary size, can quickly run into billions (Mikolov et al., 2013).

### 1.1 The Hardness Associated with the Large Number of Classes

Due to the popularity of word classification, it is not difficult to find public datasets with a large number of classes. Microsoft released ODP data (Choromanska & Langford, 2015), where the task is to predict the category for each document (see Section 4.1) which has more than 100,000 classes. The favorite Imagenet dataset, with fine-grained categories, has over 21,000 classes.

**Deployment Cost:** The computational as well as the memory requirements of the classifiers scale linearly with the number of classes $K$. For $d$ dimensional dataset the memory required by simple linear models, such as logistic regression, is $O(Kd)$, which is a significant cost for deployment.

For example, for ODP dataset, with 400,000 features and 100,000 classes, the model size of simple logistic regression is $4 \times 10^{10} \times 8$ bytes, assuming 32 bits double, or 160 gigabytes just to store the model (or parameters). Such a large model size will run out of any GPU memory, and even deploying the model size is out of the scope of the main memory of any ordinary machine. The situation naturally applied to deep networks with softmax output layer. Not surprisingly, the final layer in deep networks is a known memory bottleneck.

Furthermore, inference with linear classifier requires $K$ inner products of size $d$ each, totaling $O(dK)$ multiplications just to predict one instance. Such high inference cost is near-infeasible in many online and latency-critical applications.

**Training Cost:** Training with large parameter space, such as 40 billion in case of ODP dataset, is always an issue. The iterations are not parallelizable. With large number of parameters and sequential updates, the convergence is time consuming.

As argued, due to significant model sizes (160GB or higher) we cannot train ODP dataset on a single GPU. All existing methods use either distributed cluster or large server machines with a large amount of main memory (RAM).

## 1.2 MEMORY-COMPUTATIONS TRADEOFF: PRIOR ART

In the context of multi-class classification, reducing the prediction time at the cost of increase model size is studied recently. There are two main lines of work in this regard. It was first observed in (Shrivastava & Li, 2014) that the prediction cost, of multi-class classification problem, can be reduced to a maximum inner product search instance which admits faster solution using locality sensitive hashing. (Vijayanarasimhan et al., 2014) demonstrated this empirically in the context of deep networks. However, this reduction comes at the cost of increased memory required for storing hash tables, along with $O(Kd)$ for the parameters. (Daume III et al., 2016), instead of hash tables, uses smartly learned trees to prune down the search space quickly. However, they still requiring storing all the parameters and the additional memory needed to save trees doubles the overall space complexity. Furthermore, the parallelism is hurt in the inference time due to the tree traversals.

Memory is the primary barrier in extreme classification both during training and testing. The computation required for prediction, in one-vs-all classification, is trivially parallelizable. For example, in logistic regression, all the $K$ probabilities (inner products) can be computed in parallel followed by computing the maximum via reduction. Thus, latency can be reduced with more parallelization. However, memory creates the major hurdle in deployment and communication cost of distributed computing is prohibitive.

Unfortunately, reducing memory in extreme scale classification has not received enough formal treatment. There are, however, several tricks to reduce memory in multi-class classification. The high-level idea is to compress the parameter space, and whenever the parameters are needed, they are reconstructed back. Notable approaches include hashed nets (Chen et al., 2015), Huffman compression (Han et al., 2015), etc. As expected, the compression and decompression for accessing any parameter lead to costly overheads. Furthermore, most of the compression methods only lead to around a constant factor improvement without any formal information theoretic guarantees.

**A Note on Dimensionality Reduction:** Dimensionality reduction techniques such as random projections and feature hashing (Weinberger et al., 2009) is an orthogonal idea to reduce computations and memory by decreasing the factor $d$. On the contrary, this paper is focused on reducing the factor $K$. All the analysis and results of the paper assuming $d$ is the optimum dimensionality which is sufficient for learning.

**Difference from Multi-Label and Structured Predictions:** There are other tasks with a large number of classes such as multi-label learning (Yu et al., 2014) and structured prediction (BakIr, 2007). If we allow multiple outputs for each example, then every combination can easily give us a potentially large number of classes. However, the classes are not independent, and the structure can be exploited to reduce both computations and memory (Jasinska & Karampatziakis, 2016).

**Our Focus:** We do not make any assumptions on the classes. Our results are for any generic $K$-class classification without any relations, whatsoever, between the classes. For the sake of readability, we use standard logistic regression as a running example for our technique and experimentations. However, the results naturally extend to any $K$-class classifier, including deep networks.

*Existing methods, reduce inference time at the cost of increased memory and vice versa. It is long known that there is a tradeoff between computations, memory, and accuracy. However, in the context of multi-class classification, even with approximations, there are no principled methods that can reduce both computation and memory simultaneously, compared to $O(Kd)$, while providing theoretical guarantees of the tradeoffs. This work presents the first such result for $K$-class classification.*

### 1.3 OUR CONTRIBUTIONS:

We propose a simple hashing based divide and conquer algorithm MACH (Merged-Average Classification via Hashing) for $K$-class classification, which only requires $O(d \log K)$ model size (memory) instead of $O(Kd)$ required by traditional linear classifiers. MACH also provides computational savings by requiring only $O(d \log K + K \log K)$ multiplications instead of $O(Kd)$. Furthermore, the training process of MACH is embarrassingly parallelizable.

Overall, MACH uses a 2-universal random hash function to assign classes to a small number of buckets. The large output classification problem is reduced to a small output space problem. A small classifier is built on these merged buckets (or meta-classes). Using only logarithmic (in $K$) such independent classifiers, MACH can discriminate any pair of classes with high probability.

We provide strong theoretical guarantees quantifying the tradeoffs between computations accuracy and memory. In particular, we show that in $\log \frac{K}{\sqrt{\delta}} d$ memory MACH can discriminate between any two pair of classes with probability $1-\delta$. Our analysis provides a novel treatment for approximation in classification via a property call distinguishability between any pair of classes. Our novel formalism of approximate multi-class classification and its strong connections with compressed sensing and heavy hitters problem could be of independent interest in itself.

MACH achieves 19.28% accuracy on the ODP dataset which is the best-reported accuracy seen so far on this dataset. The best previous accuracy on this partitions was only 9% (Daume III et al., 2016). Furthermore, MACH only requires a single GPU (Nvidia Pascal Titan X). The model size with MACH is mere around 1.2GB, to achieve around 15% accuracy compared to around 160GB with the one-vs-all classifier that gets 9% and requires high-memory servers to process heavy models.

A sequential training of MACH on a single Titan X GPU takes only 7 hours on the ODP dataset. As MACH is trivially parallelizable, the same training time becomes 17 minutes with 25 GPUs. Similarly, we can train linear classifiers on fine-grained imagenet features, with 21000 categories, in around 23 hours on a single titan X and around an hour using 20 GPUs.

To the best of our knowledge, this is the first work to demonstrate training of linear classifiers over 100,000 classes on a single GPU, and the first demonstration of training fine-grained Imagenet, with 21000 classes, on a single GPU. These new results illustrate the power of MACH for extreme-classification scenarios.

## 2 BACKGROUND

We will use the standard [] for integer range, i.e., $[l]$ denotes the integer set from 1 to $l$: $[l] = \{1, 2, \cdots, l\}$. We will use the standard logistic regressions settings. The data $D$ is given by $D = (x_i, y_i)_{i=1}^{N}$. $x_i \in \mathbb{R}^d$ will be $d$ dimensional features and $y_i \in \{1, 2, \cdots, K\}$, where $K$ denote the number of categories in the multi-class classification problem. We will drop the subscript $i$ for simplicity whenever we are talking about a generic data and only use $(x, y)$. Given an $x$, we will denote the probability of $y$ (label) taking the value $i$, under the given classifier model, as $p_i = Pr(y = i|x)$.

### 2.1 2-UNIVERSAL HASHING

**Definition:** A randomized function $h : [l] \rightarrow [B]$ is 2-universal if for all, $i, j \in [l]$ with $i \neq j$, we have the following property for any $z_1, z_2 \in [k]$

$$Pr(h(i) = z_1 \text{ and } h(j) = z_2) = \frac{1}{B} \tag{1}$$

(Carter & Wegman, 1977) showed that the simplest way to create a 2-universal hashing scheme is to pick a prime number $p \geq B$, sample two random numbers $a$, $b$ uniformly in the range $[0, p]$ and compute $h(x) = ((ax + b) \bmod p) \bmod B$

## 3   MERGED-AVERAGED CLASSIFIERS VIA HASHING (MACH)

We present Merged-Averaged Classifiers via Hashing (MACH), which is the first algorithm in the classification literature that shows the power of universal hashing in reducing both computations as well as memory in extreme multi-class classification problems. Our idea is simple and comes with sound theoretical guarantees. Our procedure randomly merges classes into a small, manageable number of random-meta-classes (or buckets). We then run any off-the-shelf classifier, such as logistic regression or deep networks, on this meta-class classification problem. We then repeat the process independently $R = O(\log K)$ times each time using an independent 2-universal hashing scheme. During prediction, we aggregate the output from each of the $R$ classifiers models to obtain the predicted class. We later show that this simple scheme is theoretically sound and only needs $\log K$ memory. We present information theoretic connections of our scheme with compressed sensing and heavy hitters literature.

Formally, we use $R$, independently chosen, 2-universal hash functions $h_i : [K] \rightarrow [B]$, $i = \{1, 2, \cdots, R\}$. Each $h_i$ uniformly maps the classes (total $K$ classes) into one of the $B$ buckets. $B$ and $R$ are our parameters that we can tune to trade accuracy with both computations and memory. $B$ is usually a small constant like 10 or 50. Given the data $\{x_i, y_i\}_{i=1}^{N}$, it is convenient to visualize, that each hash function $h_j$, transforms the data $D$ into $D_j = \{x_i, h_j(y_i)\}_{i=1}^{N}$. We do not have to materialize the hashed class values for all small datasets, we can simply access the class values through $h_j$. We then train $R$ classifiers (models) on each of these $D_j$'s to get $R$ (small) models $M_j$s. This concludes our training process. Note each $h_j$ is independent. Training $R$ classifiers is trivially parallelizable across $R$ machines or GPUs.

We need a few more notations. Each meta-classifier can only classify among the merged meta-classes. Let us denote the probability of the meta-class $b \in [B]$, with the $j^{th}$ classifier with capitalized $P_b^j$. If the meta-class contains the class $i$, i.e. $h_j(i) = b$, then we can also write it as $P_{h_j(i)}^j$. We are now ready to describe the prediction phase.

During prediction, we show (in Section 3.1) that the following expression is a good estimator of the probability of class $i$, given any feature vector $x$.

$$Pr(y = i|x) \;=\; \frac{B}{B-1}\left[\frac{1}{R}\sum_{j=1}^{R} P_{h_j(i)}^j(x) - \frac{1}{B}\right], \tag{2}$$

here $P_{h_j(i)}^j(x)$ is the predicted probability of meta-class $h_j(i)$ under the $j^{th}$ model ($M_j$), for given $x$. Thus, our classification rule is given by $\arg\max_i Pr(y = i|x) = \arg\max_i \sum_{j=1}^{R} P_{h_j(i)}^j(x)$. In other words, during prediction, for every class $i$, we get the probability assigned to associated meta-class $P_{h_j(i)}^j(x)$, by the $j^{th}$ classifier on $x$. We compute the sum of these assigned probabilities, where the summation is over $j$. Our predicted class is the class with the highest value of this sum. The overall details are summarized in algorithm 1 and 2.

Clearly, the total model size of MACH is $BRd$ to store $R$ models of size $Bd$ each. The prediction cost requires $RBd$ multiplications to get meta probabilities, followed by $KR$ to compute equation 2 for each of the classes, the maximum can be computed on the fly. Thus, the total cost of prediction is $RBd + KR$.

### 3.1   ANALYSIS

**No Assumptions:** We would like to emphasize that we do not make any assumption on the classes, and we do not assume any dependencies between them. It is therefore not even clear how we can go from $K$ to $\log K$.

As noted before, we use $R$ independent $B$-class classifiers each. Classification algorithms such as logistic regression and deep networks models the probability $Pr(y = i|x) = p_i$. For example, the

famous softmax or logistic modelling uses $Pr(y = i|x) = \frac{e^{\theta_i \cdot x}}{Z}$, where $Z$ is the partition function. With MACH, we use $R$ 2-universal hash functions. For every hash function $j$, we instead model $Pr(y = b|x) = P_b^j$, where $b \in [B]$. Since $b$ is a meta-class, we can also write $P_b^j$ as

$$P_j^b = \sum_{i:h_j(i)=b} p_i; \qquad 1 = \sum_{i=1}^{K} p_i = \sum_{b \in [B]} P_j^b \quad \forall j \tag{3}$$

---

**Algorithm 1:** Train

---

**Data:** $D = (X, Y) = (x_i, y_i)_{i=1}^N$. $x_i \in \mathbb{R}^d$ $y_i \in \{1, 2, \cdots, K\}$
**Input** : $B, R$
**Output:** $R$ trained multi-class classifiers
initialize $R$ 2-universal hash functions $h_1, h2, ...h_R$
initialize $result$ as an empty list
**for** $i \leftarrow 1$ **to** $R$ **do**
    $Y_{h_i} \leftarrow h_i(Y)$
    $M_i = trainLogistic(X, Y_{h_i})$
    Append $M_i$ to $result$
**end**
**return** $result$

---

**Algorithm 2:** Predict

---

**Data:** $D = (X, Y) = (x_i, y_i)_{i=1}^N$. $x_i \in \mathbb{R}^d$ $y_i \in \{1, 2, \cdots, K\}$
**Input** : $M = M_1, M_2, ..., M_R$
**Output:** $N$ predicted labels

load $R$ 2-universal hash functions $h_1, h2, ...h_R$ used in training
initialize $P$ as a an empty list
initialize $G$ as a $(|N| * K)$ matrix
**for** $i \leftarrow 1$ **to** $R$ **do**
    $P_i = getProbability(X, M_i)$
    Append $P_i$ to $P$
**end**
**for** $j \leftarrow 1$ **to** $K$ **do**
    `/* G[:,j] indicates the jth column in matrix G                    */`
    $G[:, j] = (\sum_{r=1}^{R} P_r[:, h_r(j)])/R$
**end**
**return** $argmax(G, axis=1)$

---

With the above equation, given the $R$ classifier models, an unbiased estimator of $p_i$ is:

**Theorem 1**

$$\mathbb{E}\left[\frac{B}{B-1}\left[\frac{1}{R}\sum_{j=1}^{R} P_{h_j(i)}^j(x) - \frac{1}{B}\right]\right] = Pr\left(y = i\Big|x\right) = p_i \tag{4}$$

**Proof:** Since the hash function of universal, we can always write

$$P_{h(i)} == p_i + \sum_{j \neq i} \mathbf{1}_{h(j)=h(i)} p_j,$$

where $\mathbf{1}_{h(j)=h(i)}$ is an indicator random variable with expected value of $\frac{1}{B}$.

For a $d$ dimensional dataset (or $d$ non-zeros for sparse data), the memory required by a vanilla logistic regression model (or any linear classifier) is $O(Kd)$. $O(Kd)$ is also the computational complexity of prediction. With MACH, the computational, as well as the memory complexity, is dependent on $B$ and $R$ roughly as $O(BR + KR)$. To obtain significant savings, we want $BR$ to be significantly smaller than $Kd$. We next show that $BR \approx O(\log K)$ is sufficient for identifying the final class with high probability.

**Definition 1 Indistinguishable Class Pairs:** *Given any two classes $c_1$ and $c_2 \in [K]$, they are indistinguishable under MACH if they fall in the same meta-class for all the $R$ hash functions, i.e., $h_j(c_1) = h_j(c_2)$ for all $j \in [R]$.*

Otherwise, there is at least one classifier which provides discriminating information between them. Given that the only sources of randomness are the independent 2-universal hash functions, we can have the following lemma

**Lemma 1** *MACH with $R$ independent $B$-class classifier models, any two original classes $c_1$ and $c_2$ $\in [K]$ will be indistinguishable with probability at most*

$$Pr(\text{classes } i \text{ and } j \text{ are indistinguishable}) \leq \left(\frac{1}{B}\right)^R \tag{5}$$

There are total $\frac{K(K-1)}{2} \leq K^2$ possible pairs, and therefore, the probability that there exist at least one pair of classes, which is indistinguishable under MACH is given by the union bound as

$$Pr(\exists \text{ an indistinguishable pair}) \leq K^2 \left(\frac{1}{B}\right)^R \tag{6}$$

Thus, all we need is $K^2 \left(\frac{1}{B}\right)^R \leq \delta$ to ensure that there is no indistinguishable pair with probability $\geq 1 - \delta$. Overall, we get the following theorem:

**Theorem 2** *For any $B$, $R = \frac{2 \log \frac{K}{\sqrt{\delta}}}{\log B}$, guarantees that all pairs of classes $c_i$ and $c_j$ are distinguishable (not indistinguishable) from each other with probability greater than $1 - \delta$.*

From section 3, our memory cost is $BRd$ to guarantee all pair distinguishably with probability $1 - \delta$, which is equal to $\frac{2 \log \frac{K}{\sqrt{\delta}}}{\log B} Bd$. This holds for any constant value of $B \geq 2$. Thus, we bring the dependency on memory from $O(Kd)$ to $O(\log Kd)$ in general with approximations. Our inference cost is $\frac{2 \log \frac{K}{\sqrt{\delta}}}{\log B} Bd + \frac{2 \log \frac{K}{\sqrt{\delta}}}{\log B} K$ which is $O(K \log K + d \log K)$, which for high dimensional dataset can be significantly smaller than $Kd$.

### 3.2 Connections with Compressed Sensing and Heavy Hitters

Given a data instance $x$, a generic classifier outputs the probabilities $p_i$, $i \in \{1, 2, ..., K\}$. We can then use these $K$ numbers (probabilities) for inference and training. What we want is essentially to compress the information of these $K$ numbers to $\log K$, i.e., we can only keep track of $\log K = BR$ numbers (or measurements). Ideally, without any assumption, we cannot compress the information in $K$ numbers to anything less than $\Omega(K)$, if we want to retain all information. However, in classification, the most informative quantity is the identity of $\arg \max p_i$.

If we assume that $\max p_i \geq \frac{1}{m}$, for sufficiently small $m \leq K$, which should be true for any good classifier. Then, identifying $\arg \max p_i$ with $\sum p_i = 1$ and $\max p_i \geq \frac{1}{m} \sum p_i$ is a classical heavy hitter problem. Finding heavy hitters in the data stream is more than 30 years old problem with a rich literature. Count-Min sketch (Cormode & Muthukrishnan, 2005) is the most popular algorithm for solving heavy hitters over positive streams. Our method of using $R$ universal hashes with $B$ range is precisely the count-min sketch measurements, which we know preserve sufficient information to identify heavy hitters (or sparsity) under good signal-to-noise ratio. Thus, if $\max p_i$ (signal) is larger than $p_j$, $j \neq i$ (noise), then we should be able to identify the heavy co-ordinates (sparsity structure) in $sparsity \times \log K$ measurements (or memory) (Candes & Tao, 2006).

Our work provides the first bridge that connects compressed sensing and sketching algorithms from the data streams literature with classical extreme $K$-class classification. This connection also emphasizes that fundamental advancements in solving computational problems go way farther.

| Name | Type | #Train / #Test | #Classes | #Features | size |
|------|------|----------------|----------|-----------|------|
| ODP | Text | 1084404 / 493014 | 105033 | 422713 | 3GB(sparse format) |
| Imagenet | Images | 12777062 / 1419674 | 21841 | 6144 | 104GB(Compressed) |

Table 1: Statistics of Datasets used for Evaluations

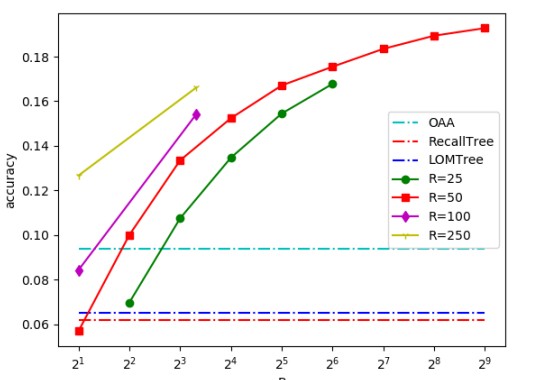 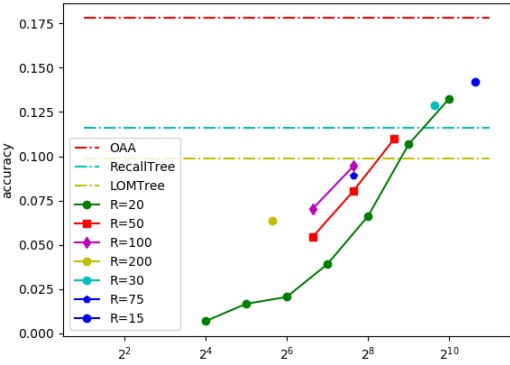

Figure 1: Accuracy Resource tradeoff with MACH (all bold lines) with varying settings of $R$ and $B$. The number of parameters are $BRd$ while the prediction time requires $KR + BRd$ operations. All the runs of MACH requires less memory than OAA. The straight line are accuracies of OAA, LOMTree and Recall Tree (dotted lines) on the same partition taken from (Daume III et al., 2016). LOMTree and Recall Tree uses more (around twice) the memory required by OAA. **Left:** ODP Dataset. **Right:** Imagenet Dataset

# 4 EVALUATIONS

## 4.1 DATASETS

We use the two large public benchmarks from (Daume III et al., 2016): 1) ODP dataset and 2) Fine-grained Imagenet.

ODP is a multi-class dataset extracted from Open Directory Project, the largest, most comprehensive human-edited directory of the Web. Each sample in the dataset is a document, and the feature representation is bag-of-words. The class label is the category associated with the document. The dataset is obtained from (Choromanska & Langford, 2015). The number of categories in this dataset is 105,033.

Imagenet is a dataset consist of features extracted from an intermediate layer of a convolutional neural network trained on the ILVSRC2012 challenge dataset. This dataset was originally developed to study transfer learning in visual tasks (Oquab et al., 2014). Please see (Choromanska & Langford, 2015) for more details. The class label is the fine-grained object category present in the image. The number of categories in this dataset is 21,841.

The statistics of the two datasets are summarized in Table 1.These datasets cover different domains – images and text. The datasets are sufficiently hard and demonstrate the challenges of having large classes. There is only one published work (Daume III et al., 2016) which could successfully train a standard one-vs-all linear classifier. The training was done over a sophisticated distributed machine and the training time spanned several days.

All the experiments are performed on the same server with GeForce GTX TITAN X, Intel(R) Core(TM) i7-5960X 8-core CPU @ 3.00GHz and 64GB memory. We used Tensorflow to train each individual model $M_i$ and obtain the probability matrix $P_i$ from model $M_i$. We use OpenCL to compute the global score matrix that encodes the score for all classes $[1, K]$ in testing data and perform argmax to find the predicted class. We have our codes and scripts ready for release on Github for reproducibility.

## 4.2 ACCURACY BASELINES

On these large benchmarks there are three published methods that have reported successful evaluations – 1) OAA, traditional one-vs-all classifiers, 2) LOMTree and 3) Recall Tree. The result of all

| Dataset | (B, R) | Model size Reduction | Training Time | Prediction Time per Query | Accuracy |
|---------|--------|---------------------|---------------|---------------------------|----------|
| ODP | (32, 25) | 125x | 7.2hrs | 2.85ms | 15.446% |
| Imagenet | (512, 20) | 2x | 23hrs | 8.5ms | 10.675% |

Table 2: Wall Clock Execution Times for two runs of MACH on a single Titan X.

these methods are taken from (Daume III et al., 2016). OAA is the standard one-vs-all classifiers whereas LOMTree and Recall Tree are tree-based methods to reduce the computational cost of prediction at the cost of increased model size. Recall Tree uses twice as much model size compared to OAA. Even LOMtree has significantly more parameters than OAA. Thus, our proposal MACH is the only method that reduces the model size compared to OAA.

We will primarily use these baselines to contrast the accuracy-memory tradeoff provided by MACH. Our system is currently not capable of running any of the above methods. For example on ODP datasets, all these methods will require more than 160GB model file, while the memory capacity of our system is 64GB only. However, since we are using the same partitions, we can directly use the reported accuracy as benchmarks to compare with.

### 4.3 RESULTS AND DISCUSSIONS

We run MACH on these two datasets varying $B$ and $R$. $B$ and $R$ are two knobs in MACH to balance resource and accuracy. We used plain logistic regression classifier, i.e., cross entropy loss without any regularization, in the Tensorflow environment (Abadi et al., 2016). We plot the accuracy as a function of different values of $B$ and $R$ in Figure 1.

The plots show that for ODP dataset MACH can even surpass OAA achieving 18% accuracy while the best-known accuracy on this partition is only 9%. LOMtree and Recall Tree can only achieve 6-6.5% accuracy. It should be noted that with 100,000 classes, a random accuracy is $10^{-5}$. Thus, the improvements are staggering with MACH. Even with $B = 32$ and $R = 25$, we can obtain more than 15% accuracy with $\frac{105,000}{32 \times 25} = 120$ times reduction in the model size. Thus, OAA needs 160GB model size, while we only need around 1.2GB. To get the same accuracy as OAA, we only need $R = 50$ and $B = 4$, which is a 480x reduction in model size requiring mere 0.3GB model file.

We believe that randomization and averaging in MACH cancel the noise and lead to better generalization. Another reason for the poor accuracy of other baselines could be due to the use of VW (Langford et al., 2007) platforms. VW platform inherently uses feature hashing that may lose some information in features, which is critical for high dimensional datasets such as ODP.

On Imagenet dataset, MACH can achieve around 11% which is roughly the same accuracy of LOMTree and Recall Tree while using $R = 20$ and $B = 512$. With $R = 20$ and $B = 512$, the memory requirement is $\frac{21841}{512 \times 20} = 2$ times less than that of OAA. On the contrary, Recall Tree and LOMTree use 2x more memory than OAA. OAA achieves the best result of 17%. With MACH, we can run at any memory budget.

### 4.4 RUNNING TIMES AND PARALLELISM

All our experiments used a single Titan X to run the combinations shown in Figure 1. There has been no prior work that has reported the run of these two datasets on a single Titan X GPU with 12GB memory. This is not surprising due to memory limit.

With MACH, training is embarrassingly easy, all we need is to run $R$ small logistic regression which is trivially parallelization over $R$ GPUs. In Table 2, we have compiled the running time of some of the reasonable combination and have shown the training and prediction time. The prediction time includes the work of computing probabilities of meta-classes followed by sequential aggregation of probabilities and finding the class with the max probability.

It should be noted that if we have $R$ machines then this time would go down by factor of $R$, due to trivial parallelism of MACH because all classifiers are completely independent. Although, we cannot compare the running time with any previous methods because they use different systems. Nonetheless, the wall clock times are significantly faster than the one reported by RecallTree, which is optimized for inference. The most exciting message is that with MACH we can train this intimidating datasets on a single GPU. We hope MACH gets adopted in practice.

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
