# OpenReview forum: "MACH: Embarrassingly parallel $K$-class classification in $O(d\log{K})$ memory and $O(K\log{K} + d\log{K})$ time, instead of $O(Kd)$"
_ICLR.cc/2018/Conference — Reject_

### Official Review · AnonReviewer3 · 2017-11-27
**Good ideas, but insufficient results**

**Rating:** 6
**Confidence:** 4

**Review:**

The manuscript proposes an efficient hashing method, namely MACH, for softmax approximation in the context of large output space, which saves both memory and computation. In particular, the proposed MACH uses 2-universal hashing to randomly group classes, and trains a classifier to predict the group membership. It does this procedure multiple times to reduce the collision and trains a classifier for each run. The final prediction is the average of all classifiers up to some constant bias and multiplier as shown in Eq (2).

The manuscript is well written and easy to follow. The idea is novel as far as I know. And it saves both training time and prediction time. One unique advantage of the proposed method is that, during inference, the likelihood of a given class can be computed very efficiently without computing the expensive partition function as in traditional softmax and many other softmax variants. Another impressive advantage is that the training and prediction is embarrassingly parallel, and thus can be linearly sped up, which is very practical and rarely seen in other softmax approximation.

Though the results on ODP dataset is very strong, the experiments still leave something to be desired.
(1) More baselines should be compared. There are lots of softmax variants for dealing with large output space, such as NCE, hierarchical softmax, adaptive softmax ("Efficient softmax approximation for GPUs" by Grave et. al), LSH hashing (as cited in the manuscript) and matrix factorization (adding one more hidden layer). The results of MACH would be more significant if comparison to these or some of these baselines can be available.
(2) More datasets should be evaluated. In this manuscript, only ODP and imagenet are evaluated. However, there are also lots of other datasets available, especially in the area of language modeling, such as one billion word dataset ("One billion
word benchmark for measuring progress in statistical language modeling" by Chelba et. al) and many others.
(3) Why the experiments only focus on simple logistic regression? With neural network, it could actually save computation and memory. For example, if one more hidden layer with M hidden units is added, then the memory consumption would be M(d+K) rather than Kd. And M could be a much smaller number, such as 512. I guess the accuracy might possibly be improved, though the memory is still linear in K.

Minor issues:
(1) In Eq (3), it should be P^j_b rather than P^b_j?
(2) The proof of theorem 1 seems unfinished

---

> ### Author Response · Authors · 2017-12-24
> **Thanks for nice comments. The methods you mentioned do not save memory**
>
> First of all, we appreciate your detail comments, spotting typos, and encouragement.
>
> (1) Hierarchical softmax and LSH does not save memory; they make memory worse compared to the vanilla classifier.
> Hierarchical softmax and any tree-like structure will lead to more (around twice) memory compared to the vanilla classifier. Every leaf (K leaves) requires memory (for a vector), and hence the total memory is of the order 2k  ( K + K/2 + ...) .  Of course, running time will be log(K).
> In theory, LSH requires K^{1 + \rho} memory ( way more than K or 2K). We still need all the weights.
> Memory is the prime bottleneck for scalability.  Note prediction is parallelizable over K (then argmax) even for vanilla models.  Thus prediction time is not a major barrier with parallelism.
>
> We stress, (to the best of our knowledge) no known method can train ODP dataset on a single Titan X with 12GB memory. All other methods will need more than 160gb main memory. The comparison will be trivial, they all will go out of memory.
>
> Also, see new comparisons with Dismec and PDsparse algorithms (similar) in comment to AnonReviewer1
>
> matrix factorization (see 3)
>
> 2) ODP is similar domain as word2vec.  We are not sure, but direct classification accuracy in word2vec does not make sense (does it?), it is usually for word embeddings (or other language models) which need all the parameters as those are the required outputs, not the class label (which is argmax ).
>
> 3) What you are mentioning (similar to matrix factorization) is a form of dimensionality reduction from D to M. As mentioned in the paper, this is orthogonal and complementary.  We can treat the final layer as the candidate for MACH for more savings.  As you said, just dimentionality reduction won't be logarithmic in K by itself.
>
>
> We thank you again for the encouragement and hope that your opinion will be even more favorable after the discussions mentioned above.

---

### Official Review · AnonReviewer1 · 2017-11-27
**Extreme multi-class classification with Hashing**

**Rating:** 6
**Confidence:** 4

**Review:**

Thanks to the authors for their feedback.
==============================
The paper presents a method for classification scheme for problems involving large number of classes in multi-class setting. This is related to the theme of extreme classification but the setting is restricted to that of multi-class classification instead of multi-label classification. The training process involves data transformation using R hash functions, and then learning R classifiers. During prediction the probability of a test instance belonging to a class is given by the sum of the probabilities assigned by the R meta-classifiers to the meta-class in the which the given class label falls. The paper demonstrates better results on ODP and Imagenet-21K datasets compared to LOMTree, RecallTree and OAA.

There are following concerns regarding the paper which don't seem to be adequately addressed :

 - The paper seems to propose a method in which two-step trees are being constructed based on random binning of labels, such that the first level has B nodes. It is not intuitively clear why such a method could be better in terms of prediction accuracy than OAA. The authors mention algorithms for training and prediction, and go on to mention that the method performs better than OAA. Also, please refer to point 2 below.

 - The paper repeatedly mentions that OAA has O(Kd) storage and prediction complexity. This is however not entirely true due to sparsity of training data, and the model. These statements seem quite misleading especially in the context of text datasets such as ODP. The authors are requested to check the papers [1] and [2], in which it is shown that OAA can perform surprisingly well. Also, exploiting the sparsity in the data/models, actual model sizes for WikiLSHTC-325K from [3] can be reduced from around 900GB to less than 10GB with weight pruning, and sparsity inducing regularizers. It is not clear if the 160GB model size reported for ODP took the above suggestions into considerations, and which kind of regularization was used. Was the solver used from vowpal wabbit or packages such as Liblinear were used for reporting OAA results.

 - Lack of empirical comparison - The paper lacks empirical comparisons especially on large-scale multi-class LSHTC-1/2/3 datasets [4] on which many approaches have been proposed. For a fair comparison, the proposed method must be compared against these datasets. It would be important to clarify if the method can be used on multi-label datasets or not, if so, it needs to be evaluated on the XML datasets [3].

[1] PPDSparse - http://www.kdd.org/kdd2017/papers/view/a-parallel-and-primal-dual-sparse-method-for-extreme-classification
[2] DiSMEC - https://arxiv.org/abs/1609.02521
[3] http://manikvarma.org/downloads/XC/XMLRepository.html
[4] http://lshtc.iit.demokritos.gr/LSHTC2_CFP

---

> ### Author Response · Authors · 2017-12-24
> **MACH seems superior (more experiments)**
>
> Thanks for pointing our sparsity and also reference related.  We tried compared with [1] and [2] (referred in your comment) as pointed out on ODP dataset,  and we are delighted to share the results.  We hope these results (below) will convince you that
> 1) we are indeed using challenging large-scale dataset.
> 2) sparsity is nice to reduce the model size, but training is prohibitively slow. We still have 40 billion parameters to think about, even if we are not storing all of them (See results of dismec)
> 3) And our proposal is blazing fast and accurate and above all simple. Afterall what will beat small (32 classes only instead of 100k) logistic regression?
> 4) Still, we stress, (to the best of our knowledge) no known method can train ODP dataset on a single Titan X.
>
> We will add the new results in any future version of the paper.
>
> First of all, ODP is a large scale dataset, evident from the fact that both the methods [1] and [2] are either prohibitively slow or goes out or memory.
>
> It is perfectly fine to have sparse models which will make the final model small in memory. The major hurdle is to train them.  We have no idea which weights are sparse. So the only hope to always keep the memory small is some variant of iterative hard thresholding to get rid of small weights repetitively.  That is what is done by
> Dismec, reference [2].  As expected, this should be very slow.
>
> ****** Dismec Details on ODP dataset***********
>
> We tried running dismec with the recommended control model set.
> Control Model size: Set a ambiguity control hyper-parameter delta (0.01). if a value in weight matrix is between -delta and delta, prune the value because the value carries very little discriminative information of distinguishing one label against another.
>
> Running time: approx. 3 models / 24h, requires 106 models for ODP dataset, approx. 35 days to finish training on Rush. We haven't finished it yet.
> Compare this to our proposed MACH which takes 7.3 hrs on a single GPU.  Afterall, we are training small logistic regression with 32 classes only, its blazing fast. No iterative thresholding, not slow training.
>
> Furthermore, Dismec does not come with probabilistic guarantees of log{K} memory. Sparsity is also a very specific assumption and not always the way to reduce model size.
>
> The results are not surprising as in [2] sophisticated computers with 300-1000 cores were used.  We use a simple machine with a single Titan X.
>
> ********** PD-Sparse**************
>
> We also ran  PD-sparse a non-parallel version [1] (we couldn't find the code for [1]), but it should have same memory consumption as [1]. The difference seems regarding parallelization.  We again used the ODP dataset with recommended settings.   We couldn't run it. Below are details
>
> It goes out of memory on our 64gb machine.  So we tried using another 512GB RAM machine, it failed after consuming 70% of memory.
>
> To do a cross sanity check, we ran PPD on LSHTC1 (one of the datasets used in the original paper [1]). It went out of memory on our machine (64 GB) but worked on 512 GB RAM Machine with accuracy as expected in [1].  Interestingly, the run consumed more than 343 GB of main memory.  This is ten times more than the memory required for storing KD double this dataset with K =12294 and D=347255.
> ***********************************
>
> Let us know if you are still not convinced. We are excited about MACH, a really simple, theoretically sound algorithm, for extreme class classification.  No bells and whistles,  no assumption, not even sparsity.

---

> > ### Comment · AnonReviewer1 · 2017-12-26
> > **DiSMEC on ODP dataset**
> >
> > Thanks for the update on various points.
> >
> > I would disagree with some of the responses particularly on sparsity, on the merit of using a single Titan X and hence the projected training time mentioned for DiSMEC on ODP dataset. These are mentioned in detail below. Before that I would like to mention some of my empirical findings.
> >
> > To verify my doubts on using DiSMEC on ODP as in the initial review, I was able to run it in a day or so, since I had access to a few hundreds cores. It turns out it gives an accuracy of 24.8% which is about 30% better than MACH, and much better than reported for the OAA performance in earlier papers such as Daume etal [1] which reported 9% on this dataset.
> >
> > Furthermore, after storing the model in sparse format, the model size was around 3.1GB, instead of 160 GB as mentioned in this and earlier papers. It would be great if the authors could verify these findings if they have access to a moderately sized cluster with a few hundred cores. If the authors then agree, it would be great to mention these in the new version of the paper for future references.
> >
> >  - Sparsity : For text dataset with large number of labels such as in ODP, it is quite common for the model to be sparse. This is because, all the words/features are highly unlikely to be surely present or surely not present for each label/class. Therefore, there is bound to lots of zeros in the model. From an information theoretic view-point as well, it does not make much of a sense for ODP model to be 160GB when the training data is 4GB. Therefore, sparsity is not merely an assumption as an approximation but is a reasonable way to control model complexity and hence the model size.
> >
> > - Computational resources - The argument of the paper mainly hinges on the usage of a single Titan X. However, it is not clear what is the use-case/scenario in which one wants to train strictly on a single GPU. This needs to be appropriately emphasized and explained.  On the other hand, a few hundred/thousands cores is something which typically is available in organizations/institutions which might care about problems of large sizes such as on ODP and Imagenet dataset.
> >
> > Also, the authors can download the PPDSparse code from XMC respository or directly from the link http://www.cs.cmu.edu/~eyan/software/AsyncPDSparse.zip
> >
> > [1] Logarithmic Time One-Against-Some, ICML 2017

---

> > > ### Author Response · Authors · 2017-12-26
> > > **Memory usage, time ? (We want to add this comparison to the paper)**
> > >
> > > We are really grateful for your efforts and taking time to run dismec.
> > > Could you send us details of (or link to your codes?). We would like to report this in the paper (and also the comparison with imagenet).
> > > We want to know memory usage, running time (approx a day?), how many cores.  In our codes, dismec was run on a single 64GB machine, with 8 cores and one titan X.
> > >
> > > Furthermore, on imagenet, sparsity won't help. MACH does not need this assumption.  So we need to think beyond sparsity.
> > >
> > > MACH has all these properties.
> > >
> > > The main argument is that we can run on Titan X (<  12GB working memory) (sequentially run 25 logistic regression of size 32 classes each) in 7.2 hrs.  If we run with 25 GPUs in parallel, then it can be done in 17 minutes!  Compare this to about a day on a large machine.
> > >
> > > We think the ability to train dataset on GPUs or even single GPU is very impactful. GPU clusters are everywhere and cheap now.  If we can train in few hours on easily available single GPU or in few minutes on 25 GPUs (also cheap to have).  Then why wait over a day on a high-memory, high-core machines (expensive).  Furthermore, with data growing faster than our machines, any work which enhances our capability to train them is beneficial.
> > >
> > > We hope you see the importance of simplicity of our method and how fast we can train with increased parallelism. 17 min on 25 Titan X. The parallelism is trivial.
> > >
> > > We are happy to run any specific benchmark (head-to-head) you have in mind if that could convince you.

---

> > > > ### Comment · AnonReviewer1 · 2017-12-26
> > > > **results**
> > > >
> > > > Thanks for your feedback.
> > > >
> > > > The results above were for DiSMEC and not PPDSparse. Since it trains One-versus-rest in a parallel way, the memory requirements on a single node are quite moderate, something around 8GB for training a batch of 1,000 labels on a single node. Each batch of labels is trained on a separate node.
> > > >
> > > > You are absolutely right that sparsity does not make sense in case of Imagenet and the results for OAA in Figure 1(right) will hold. In both cases OAA seems to be better than MACH.
> > > >
> > > > I completely agree that MACH has computational advantages. However, at the same time, the performance is also lost in the speedup gain, i.e. 25% versus 19%. The impact of MACH would be substantial if similar levels of accuracy at much lower computational cost.
> > > >
> > > > It is important that authors could verify these, and update the manuscript appropriately thereby mentioning the pros and cons of each scheme, which is missing from the current version.

---

> > > > > ### Author Response · Authors · 2018-01-04
> > > > > **We will add the required discussion**
> > > > >
> > > > > Alright, we will add a discussion about dismec and ppd-sparse and also make a note about this conversation, and the results over 100 node machine. It seems sparsity regularizer can help with accuracy (known in literature). We note that MACH does not use any such regularization.
> > > > >
> > > > > Till now, we were still trying to run DisMEC using our machines which we can access (56 cored and 512 RAM) on both the datasets Imagenet and ODP. However, the results seem hopeless so far, and it seems the progress is significantly slower on both of them.  It will take a couple of weeks more before we can see the final accuracy (if the machines don't crash).  Imagenet seems even worse. It would be a lot of more convenient to report some official accuracy numbers if we can get them.
> > > > >
> > > > > It should be noted that we can run MACH on both the dataset over a smaller and significantly cheaper machine (64GB and 1 Titan X with 8 cores) and in substantially lesser time.
> > > > >
> > > > > We thank you for bringing these newer comparisons. We think it makes our method even more exciting and bolsters our arguments further.
> > > > >
> > > > > We hope that under the light of these discussions you will be more supportive of the paper.
> > > > >
> > > > > We will be happy to take into account any other suggestions.
> > > > >
> > > > > Thanks again, we appreciate your efforts, and we find this discussion very useful.

---

### Official Review · AnonReviewer2 · 2017-11-27
**MACH: Embarrassingly parallel $K$-class classification**

**Rating:** 6
**Confidence:** 4

**Review:**

The paper presents a hashing based scheme (MACH) for reducing memory and computation time for K-way classification when K is large. The main idea is to use R hash functions to generate R different datasets/classifiers where the K classes are mapped into a small number of buckets (B). During inference the probabilities from the R classifiers are summed up to obtain the best scoring class. The authors provide theoretical guarantees showing that both memory and computation time become functions of log(K) and thus providing significant speed-up for large scale classification problems. Results are provided on the Imagenet and ODP datasets with comparisons to regular one-vs-all classifiers and tree-based methods for speeding up classification.

Positives
- The idea of using R hash functions to remap K-way classification into R B-way classification problems is fairly novel and the authors provide sound theoretical arguments showing how the K probabilities can be approximated using the R different problems.
- The theoritical savings in memory and computation time is fairly significant and results suggest the proposed approach provides a good trade-off between accuracy and resource costs.

Negatives
- Hierarchical softmax is one of more standard techniques that has been very effective at large-scale classification. The paper does not provide comparisons with this baseline which also reduces computation time to log(K).
- The provided baselines LOMTree, Recall Tree are missing descriptions/citations. Without this it is hard to judge if these are good baselines to compare with.
- Figure 1 only shows how accuracy varies as the model parameters are varied. A better graph to include would be a time vs accuracy trade-off for all methods.
- On the Imagenet dataset the best result using the proposed approach is only 85% of the OAA baseline.  Is there any setting where the proposed approach reaches 95% of the baseline accuracy?

---

> ### Author Response · Authors · 2017-12-24
> **Thanks for Positive Feedback**
>
> Thanks for taking the time in improving our work.
>
> - We DID compare with log(K)running time methods (both LOMTree and RecallTree are log(K) running time not memory).  Hierarchical softmax and any tree-like structure will lead to more (around twice) memory compared to the vanilla classifier. Every leaf (K leaves) requires a memory and hence the total memory is of the order 2k  ( K + K/2 + ...) .  Of course running time will be log(K).
> However, as mentioned memory is prime bottleneck in scalability. We still have to update and store those many parameters.
> - Although, we have provided citations. we appreciate you pointing it out. We will make it more explicit at various places.
> - We avoided the time tradeoff because time depends on several factors like parallelism, implementation etc. For example, we can trivially parallelize across R processors.
> -  It seems there is a price for approximations on fine-grained imagenet. Even recalltree and LOMTree with twice the memory does worse than MACH.
>
> We thank you again for the encouragement and hope that your opinion will be even more positive after these discussions.

---

### Decision · Program_Chairs · 2018-01-29
**ICLR 2018 Conference Acceptance Decision**

**Decision:**

Reject

**Comment:**

There is a very nice discussion with one of the reviewers on the experiments, that I think would need to be battened down in an ideal setting. I'm also a bit surprised at the lack of discussion or comparison to two seemingly highly related papers:

1. T. G. Dietterich and G. Bakiri (1995) Solving Multiclass via Error Correcting Output Codes.
2. Hsu, Kakade, Langford and Zhang (2009) Multi-Label Prediction via Compressed Sensing.